# Genetic Variants of the NF-κB Pathway: Unraveling the Genetic Architecture of Psoriatic Disease

**DOI:** 10.3390/ijms222313004

**Published:** 2021-11-30

**Authors:** Rubén Queiro, Pablo Coto, Leire González-Lara, Eliecer Coto

**Affiliations:** 1Rheumatology & ISPA Translational Immunology Division, Hospital Universitario Central de Asturias, 33011 Oviedo, Spain; 2Department of Medicine, Oviedo University School of Medicine, 33011 Oviedo, Spain; eliecer.coto@sespa.princast.es; 3Dermatology Division, Hospital Vital Alvarez Buylla, 33611 Mieres, Spain; pablocotosegura@gmail.com; 4Dermatology Division, Hôpital Ambroise-Paré, 92100 Boulogne-Billancourt, France; leiregonzalezlara@gmail.com; 5Molecular Genetics Unit, Hospital Universitario Central Asturias, 33011 Oviedo, Spain

**Keywords:** psoriasis, psoriatic arthritis, NF-κB, comorbidities, genetic architecture

## Abstract

Psoriasis is a multifactorial genetic disease for which the genetic factors explain about 70% of disease susceptibility. Up to 30–40% of psoriasis patients develop psoriatic arthritis (PsA). However, PsA can be considered as a “disease within a disease”, since in most cases psoriasis is already present when joint complaints begin. This has made studies that attempt to unravel the genetic basis for both components of psoriatic disease enormously difficult. Psoriatic disease is also accompanied by a high burden of comorbid conditions, mainly of the cardiometabolic type. It is currently unclear whether these comorbidities and psoriatic disease have a shared genetic basis or not. The nuclear factor of kappa light chain enhancer of activated B cells (NF-κB) is a transcription factor that regulates a plethora of genes in response to infection, inflammation, and a wide variety of stimuli on several cell types. This mini-review is focused on recent findings that highlight the importance of this pathway both in the susceptibility and in the determinism of some features of psoriatic disease. We also briefly review the importance of genetic variants of this pathway as biomarkers of pharmacological response. All the above may help to better understand the etiopathogenesis of this complex entity.

## 1. Introduction

Psoriasis is a chronic immune-mediated inflammatory dermopathy that affects 2–3% of the general population. Its most common companion is psoriatic arthritis (PsA), a chronic arthritis included under the spondyloarthritis concept that affects one-third of psoriasis patients [1]. According to recent estimates, PsA can affect almost 0.6% of the adult population [2]. Both psoriasis and PsA are the main poles of what is now regarded as psoriatic disease, a systemic nature entity, where apart from the skin and musculoskeletal condition, there are a wide variety of comorbidities, the most relevant being those of cardiometabolic type [1,3]. In fact, we know of the tight connections between the inflammatory burden of psoriatic disease and higher cardiovascular risk [1,4].

Psoriatic disease itself, as well as its pleomorphic clinical manifestations, result from complex stochastic interactions between elements of genetic predisposition, immunopathological alterations, and environmental factors [1]. In the case of PsA, it is currently thought that the genetic substrate of the disease favors certain alterations of the gut microbiome of these patients, which gives rise to an IL17-type immune response, which have the potential capacity to migrate to distant sites such as entheses and joints, where the activation of certain lineages of innate cellularity (ILC-3 and γδ T cells), responsible for the beginning of the manifestations of the disease, would prevail (gut–joint axis theory) [1,5]. Adaptive immunity would act, in this case, as positive feedback for the primal responses of innate immunity [1,5].

Psoriasis is a multifactorial genetic disease for which the genetic factors explain about 70% of disease susceptibility [1,6,7]. However, PsA can be considered as a “disease within a disease”, since in most cases psoriasis is already present when arthritis begins [1]. This has made studies that attempt to unravel the genetic basis for both components of psoriatic disease enormously difficult. Moreover, this information is essential as it would help predict which psoriasis patients are at increased risk of developing arthritis over time. Genetic factors are evident from the high prevalence of PsA among first-degree relatives of PsA probands, and a recurrence risk ratio of 30–35 [1,6,7]. Also, the psoriatic disease concordance rate for monozygotic twins is higher compared to that of dizygotic twins [1]. Like psoriasis, both dominant and recessive inheritance, as well as an excessive paternal transmission pattern, has been proposed for PsA but neither apply, thus PsA has also been considered a multifactorial polygenetic disease [1].

Genome wide association studies (GWAS) have revealed more than 80 risk loci associated with psoriatic disease that are basically incorporated into three fundamental networks: (i) genes involved in the maintenance of the cutaneous barrier function, (ii) genes that control innate immune responses mediated by NF-κB and interferon signaling, (iii) genes that control adaptive immune responses that involves CD8 lymphocytes and Th17 signaling [1,7]. However, most of them are related to psoriasis risk, while only a few seem specific to PsA. Genes with genome-wide significance for PsA include *HLA-B/C, HLA-B, IL12B, IL23R, TNP1, TRAF3IP3*, and *REL* [1,7]. Although recent GWAS studies have identified numerous risk loci of psoriatic disease, almost 50% of the heritability of the disease remains unknown, and most of the genes identified so far have a small weight in the genetic etiology of the disease. This “lost” heritability can be attributed, among others, to common variants that have a very small weight in risk, copy number variants, epigenetic or epistatic interactions, or lack of power of current detection tools [1,7].

As we have commented so far, the concept of psoriatic disease not only refers to the skin or musculoskeletal domains of the disease, but also to the wide variety of comorbidities that accompany it [1,3]. Therefore, it would be interesting to know if this complex network of manifestations presents common or different genetic restriction elements. This knowledge could be key in the future to lay the foundations of truly personalized medicine for these patients.

This mini-review will focus on the importance of the NF-κB pathway, both in the predisposition to psoriatic disease, as well as in the differentiation of the genetic architecture of its different components, including the determinism of some of its comorbidities.

## 2. The NF-κB Signaling Pathway

The nuclear factor of kappa light chain enhancer of activated B cells (NF-κB) is a transcription factor that regulates a plethora of genes in response to infection, inflammation, and a wide variety of stimuli on immune cells [8,9]. This transcription factor represents a family of structurally related proteins (p100 or NFκB2, p105 or NFκB1, p65 or RelA, RelB, or c-Rel), which exist as homo- or heterodimers [8,9]. Effects of NF-κB are mediated through three pathways. In the canonical pathway, (i) phosphorylation of inhibitory IκB proteins (IkBα) leads to release of NF-κB and its nuclear translocation to promote inflammation and cell survival. The p105 pathway (ii) is dependent on phosphorylation of p105 proteins, leading to nuclear translocation of p52 heterodimer complexes to promote inflammation. Different to the above two pathways, the alternate p100 pathway (iii) does not depend on the NF-κB essential modulator (NEMO)-IKKa-IKKb (NEMO-IKK) complex for phosphorylation, but rather on NF-κB inducing kinase (NIK), and IKKa heterodimers phosphorylate p100 and allows nuclear translocation of p52/RelB heterodimers [8,9].

The pathogenetic role of NF-κB has been demonstrated in many diseases including cancer, immune-mediated inflammatory diseases (IMIDs), as well as cardiovascular diseases [9]. Genome-wide association studies have revealed several psoriatic disease susceptibility genes associated with the NF-κB pathway [10]. In addition, the role of NF-κB in psoriasis is supported by studies that reported its differential expression in normal vs. affected skin, but also by studies that showed that the inhibition of this transcriptional factor by several compounds ameliorated skin inflammation [11,12].

Due to its relevance to this review, we will refer in detail to a specific pathway, which involves TNF-α and the NF-κB transcription complex (Figure 1) with special focus on genetic variations in *CARD14*, *TNFSRF1B*, *NFKB1*, *NFKBIA*, and *NFKBIZ*.

Activation of the NF-κB pathway leads to increased transcription of numerous genes including proinflammatory cytokines, chemokines, and growth factors, all of which are involved in the onset and perpetuation of the inflammatory response in psoriatic disease [11,12]. Interestingly, the activation of NF-κB induces the production of TNF-α, which in turn can stimulate this pathway, thus resulting in positive proinflammatory feedback, and playing a key role in the chronicity of the skin/joint inflammatory response [11,12]. Moreover, an association between high levels of TNF-α and the activation of NF-κB has been found in the skin of patients with psoriasis; and therefore, a possible mechanism of action of anti-TNFα drugs would be, at least in part, the inhibition of the transcriptional activity of NF-κB [13], as we will see later.

## 3. Role in the Etiology of Psoriatic Disease

The caspase recruitment domain family member 14 (CARD14) is an intracellular scaffold protein that is prominently expressed in keratinocytes and mediates NF-κB activation through the formation of a CBM (CARD14-BCL10-MALT1) signaling complex [14]. It has been shown that gain-of-function mutations of *CARD14* enhance CBM complex formation in keratinocytes driving hyperactivation of the NF-κB pathway [15]. This leads to transcription of several chemokines (CCL20, CXCL1, and CXCL2), cytokines (IL-36 and IL-19), and antimicrobial peptides, followed by recruitment and activation of neutrophils, dendritic cells, and T cells [15]. Activated dendritic cells in turn release IL-23, which drives the downstream expression of IL-17 and IL-22, two central cytokines in the pathogenesis of the disease [15]. Variations in the *CARD14* gene have been associated with psoriatic disease. Jordan et al. identified 15 nucleotide variants in the *CARD14* gene significantly associated with the risk of psoriasis [16]. In a meta-analysis, they confirmed the association between rsll652075 (p. Arg820Trp) and psoriasis [17]. This association was later confirmed by two large meta-analysis [18,19]. We also found a statistically significant association between rsll652075 (CC genotype) and the risk of developing psoriasis (OR 1.59), regardless of the Cw6 genotype. However, we did not find genotypic differences related to psoriasis severity, age at disease onset (above or below 40 years), PsA, or family history of psoriasis [20].

The *TNFSRF1B* gene encodes the type 2 membrane receptor for TNF-α. The SNP TNFRSF1B rs1061622 (p. Met196Arg) has been implicated in the risk of several IMIDs [21,22]. This variant has hardly been studied in psoriatic disease. In our population, we did not find differences in allelic and genotypic frequencies between patients and controls for the SNP rs1061622. Neither were differences found in psoriasis severity, nor between the presence or absence of PsA. However, we did find an increased frequency of the G allele in the Cw6 + patients (OR 1.69), which suggests an increased risk of developing psoriasis conditioned by the HLA-Cw6 allele [23].

The *NFKB1* gene encodes the p50 protein, responsible for binding to the consensus sequence of the promotor region of many genes [8,9]. The most studied and best characterized polymorphism of this gene is a 4 nucleotide indel in the promoter region (rs28362491; -94 of the ATTG). The copies of the gene with the deletion would have less promoter activity and, therefore, translate into lower protein levels, which could explain the genetic associations described for this variant [24]. Other variants located at the chromosomal region where *NFKB1* is found have been investigated and related to the risk of psoriasis, specifically rs1020760 and rs1609798 [25]. A study in a Han Chinese population has recently established the association between the SNP rs1020760, psoriasis, and a positive family history of psoriasis [26]. We studied the SNP rs230526 A/G in the *NFKB1* gene, which is in complete linkage disequilibrium (LD) with rs28362491 (-94 of ATTG). Therefore, the genotypes of rs230526 would allow us to infer those of rs28362491. Despite this, we did not find any difference between allelic or genotypic frequencies in patients and controls. We also did not see a significant effect on its severity, PsA, or the age of onset of psoriasis; nor differences in relation to the presence of HLA-Cw6 [27].

*NFKBIA* gene encodes IκBα, a cytoplasmic inhibitor of the NF-κB transcriptional complex. That is, its binding to the complex keeps it inactive in the cytoplasm by blocking its ability to move to the cell nucleus. In response to certain stimuli, IκBα is phosphorylated, and this releases it from the complex, which can now migrate to the nucleus and act as a multigenic transcription factor [8,9]. Variants in this gene have been associated with the risk of myocardial infarction and several IMIDs [28,29]. In 2010, two GWAS established the association of two *NFKBIA* SNPs and the risk of psoriasis, while in 2015 a third GWAS confirmed the association of an additional SNP with psoriasis [30,31,32]. We have studied the rs7152376 SNP, which is in complete LD with rs12883343, a SNP related in the literature with both psoriasis and PsA. We did not find any association between rs7152376 and the risk of psoriasis, psoriasis severity, or the age at disease onset. However, we did find a statistically significant association between the rare G allele and the risk of PsA, with a dominant effect (AG + GG vs. AA). The rare NFKBIA rs7152376 C was significantly more frequent in PsA vs. controls (OR 2.03). The difference was even higher between PsA vs. “pure” psoriasis (OR 3.2) [33]. These findings are in line with the results of a meta-analysis that described the association of rs12883343, in complete LD with rs7152376, and the risk of PsA [34]. In this meta-analysis, the rare allele was also more frequent in PsA than in controls (meta-OR 1.22). Therefore, this variation in *NFKBIA* could be a true risk marker for PsA within the psoriatic complex.

The *NFKBIZ* gene encodes the IKBζ protein. The activity of NF-κB is regulated by nuclear inhibitors, such as IKBζ, which block its binding to DNA [8,9]. On the other hand, these peptides have a dual effect since by themselves they can act as transcription factors activating the expression of genes regulated by IL-17 or TNFα [35]. In 2015, Tsoi et al. described the genetic association between the *NFKBIZ* SNP rs7637230 and psoriasis [36]. This SNP is located outside the gene at the 3 ‘end, so it could be an indirect marker of some variant in *NFKBIZ* associated with its expression and/or function. The best candidate described so far would be an indel-type polymorphism in intron 10, rs3217713. Given its proximity to the 5 ‘end, it could influence the processing of total RNA into messenger RNA, an aspect that should be further investigated in vitro. We have investigated this indel SNP in psoriatic disease. When studying rs3217713, we did not find significant differences between patients and controls, and between the different clinical groups of patients, except for Cw6. Interestingly, the risk of developing psoriasis conferred by the combination of Cw6 + and rs3217713 ins/ins was statistically higher (OR 3.61) than the risk conferred by each genotype separately, which suggests some mechanism of genetic interaction or epistasis between both loci [37].

## 4. NF-κB Pathway Variants and Drug Response

Due to the various levels of regulation, the NF-κB signaling pathway can be potentially targeted at various levels including kinases, phosphatases, ubiquitination, nuclear translocation, DNA binding, protein acetyl transferases, and methyl transferases [38]. Currently available drugs for the treatment of psoriasis and PsA interact in some way with these targets within the NF-kB signaling pathway. For example, NSAIDs are commonly used in the treatment of PsA and have been shown to inhibit the activity of IKKβ, preventing the degradation of IκB and blocking NF-κB nuclear translocation. Glucocorticoids are highly effective in the topical treatment of active psoriatic lesions but also as systemic agents in PsA and have been shown to inhibit NF-κB through both indirect and direct mechanisms. Indirectly, glucocorticoids induce the transcription and synthesis of IκBα, enhancing the retention of NF-κB in the cytoplasm and effectively inhibiting its activation. However, under certain conditions, glucocorticoids can directly inhibit activated NF-κB via competition between p65 and the glucocorticoid receptor for limited nuclear supplies of coactivator proteins. Both receptors require these coactivators for maximal activity, and by sequestering the available stores, glucocorticoids interfere with p65-dependent gene activation. Stimulation of human synovial cells with TNFα results in NF-κB activation and subsequent cell proliferation, with NF-κB blockade able to activity inhibit this TNFα-induced proliferation. Anti-TNFα biologics effectively remove these upstream activator stimuli, and as such, the inhibition of NF-κB activation can be considered part of their mechanism of action [38]. Secukinumab, an anti–IL-17A antibody, mediates some of its antipsoriatic effects by rapidly inhibiting IκBζ and subsequently IκBζ signature genes, which highly suggests that IL-17A/IκBζ signaling is a key driver of the complex psoriatic phenotype. These data strongly indicate that an important and very early mechanism of action of anti–IL-17A therapy in patients with psoriasis is a reduction in IκBζ expression and a concomitant reduction in expression of IκBζ signature genes. It is possible that the IL-17A/IκBζ signaling axis also plays a role in the pathogenesis of psoriatic arthritis [39].

The search for genetic markers of good/bad response to drugs is a booming field, but still in its infancy. In this case, we will focus on the associations between NF-κB pathway variants and the response to anti-TNFα drugs. Anti-TNFα blocking agents could negatively modulate the signaling of the NF-κB pathway [13]. Genetic variations in the components of this pathway, particularly those that are related to the risk of psoriatic disease, could serve as pharmacogenetic markers. With respect to *CARD14* gene, we did find a significant association between rsll652075 (CC genotype) and the response to anti-TNFα drugs measured as PASI 75 at 24 weeks (OR 3.71) [40]. In our study cohorts, carriers of the G allele (GT + GG genotypes) of rs1061622 *TNFRSF1B* were associated with a worse response to anti-TNFα drugs (OR 2.34) [23]. However, the literature shows disparate results both in psoriatic disease and in other IMIDs, in this regard [41]. In our experience with anti-TNFα drugs, we did not observe any association between the genetic variants in *NFKB1*, *NFKBIA*, and drug response. However, in the case of *NFKBIZ*, the insertion allele was found more frequently in the group of non-responders, with a recessive effect (II vs. ID + DD *p* = 0.02), and OR for the negative drug response of 3.01 (95%CI: 1.15–7.88) [42]. However, the results of the literature in this regard are also disparate [43]. Anyway, it must be considered that this insertion allele is associated with the risk of psoriasis conditioned by the presence of HLA-Cw6, and the latter has also been linked to drug response in this disease [44].

## 5. NF-κB Genetic Variants and Cardiometabolic Comorbidity

Psoriatic disease is accompanied by extracutaneous and extra-articular manifestations, as well as by a wide range of comorbidities. Compared with controls, patients with psoriatic disease have a higher incidence rate of other autoimmune diseases, cardiovascular disease, obesity/overweight, depression, anxiety, smoking, cancer, diabetes, alcohol abuse, osteoporosis, uveitis, and fatty liver disease [45]. One of the common connectors between the disease and its comorbidities is inflammation. For example, chronic inflammation in both psoriatic disease and atherosclerosis promotes increased production of adipokines and pro-inflammatory cytokines (e.g., TNF) with consequent insulin resistance and endothelial dysfunction. Also, data from animal models and human studies highlight the proatherogenic role of IL-17 on vascular inflammation, acting in synergy with TNF and IL-6. IL-17 is involved in endothelial dysfunction, hypertension, plaque progression and destabilization, stroke, and myocardial infarction. IL-17 also links inflammation with insulin resistance and adipocytes dysfunction (two common aspects of the so-called psoriatic metainflammation). In mice models with systemic inflammation and insulin resistance, there is accumulation of IL-6/IL-17 co-expressing T cells in the adipose tissue; these cells enhance leptin production, which eventually acts in synergy with IL-6 and IL-17 to promote Th17 differentiation. This complex network drives local and systemic insulin resistance, and, in fact, IL-17 neutralization improves glucose uptake. Another example of these relationships is seen in osteoporosis (OP), another common comorbidity among psoriatic patients. Cytokines involved in PsA pathogenesis have an effect on bone cell activity with possible inhibitory or stimulatory stimuli on osteoclasts and osteoblasts. TNF, IL-6, and IL-1, for example, exert stimulatory activity toward osteoclasts and inhibitory activity toward osteoblasts. IL-12 and IL-23 are also critical to inflammation-induced bone resorption. Specifically, IL-23 upregulates the receptor activator of nuclear factor kappa- B (RANK) on preosteoclasts and induces Th17 cells to produce IL-17. IL-17, one of PsA signature cytokines, promotes bone resorption via RANK ligand upregulation. Indeed, Th17 cells have been found to be highly increased in blood and tissues of patients with OP [45].

In any case, for the purposes of this review, we will focus on the most important comorbidity in these patients, such as cardiovascular disease. Although psoriatic disease is frequently accompanied by cardiovascular comorbidity, the potential genetic connections between this comorbidity and psoriatic disease have received little attention. Furthermore, we do not know whether the genetic pathways to both cardiovascular comorbidity and psoriatic disease are common or differ. Although some studies suggest shared genes between psoriasis and cardiometabolic comorbidity, others suggest that the genetic architecture of psoriatic disease and cardiometabolic traits are largely distinct [46,47]. The NF-κB pathway might play a role in the pathogenesis of renal disease and type 2 diabetes mellitus (T2DM). In that sense, we studied 487 individuals, all Caucasian and aged 65–85 years. A total of 104 (21%) had impaired renal function and 30% were diabetics. The NFKB1 variants were significantly associated with T2DM: rs7667496 (OR 1.68) and rs28362491 (OR 1.67). There was a trend toward the association of these variants with impaired renal function. The two NFKB1 variants were in LD, and homozygous for the two non-risk alleles (rs7667496 CC + rs28362491 II), were significantly more common in the non-diabetics [48]. Therefore, *NFKB1* gene variations may be related to T2DM and an impaired renal function, both aspects clearly present in psoriatic patients [1,3].

Several reports show that the transcription factor NF-κB also activates genes involved in various cardiovascular diseases, in the pathogenesis of cardiac remodeling, and heart failure [49]. NF-κB is activated in the heart in many conditions: during acute ischemia and reperfusion, during unstable angina or in response to preconditioning [49]. An association between inflammation and cardiovascular risk has been suggested by the evidence that inflammatory cytokines, including IL-1β, IL-6, IL-18, and TNF-α, are increased in patients with heart failure, and increased inflammatory markers, such as C-reactive protein, predict a worse survival during acute coronary syndromes [49]. The association between psoriatic disease and cardiovascular risk has been clearly established over the last decade [1,3,4]. We determined whether common variants in three NF-κB genes were associated with early-onset coronary artery disease (CAD). Allele and genotype frequencies for the NFKB1 rs28362491 (-94 delATTG) and NFKBIA rs8904 were not significantly different between patients and controls. For the NFKBIZ rs3217713, the deletion allele was significantly more frequent in early-onset CAD patients. Deletion-carriers were more frequent in CAD (OR 1.48). Therefore, NFKBIZ variant was an independent risk factor for developing early-onset CAD [50]. However, in a recent meta-analysis of 13 case-control studies with 17 individual cohorts containing 9378 cases and 10,738 controls, the mutant D allele in NFKB1 rs28362491 locus increased the risk of CAD [51].

Table 1 summarizes the main results of this review.

Obesity, the core element of metabolic syndrome (MetS), is common among patients with psoriatic disease [1,3]. The NF-κB pathway is involved in the low-grade chronic inflammation associated with obesity. The noncanonical IkB kinase ε and TBK1 are upregulated by overnutrition and may therefore be suitable potential therapeutic targets for MetS [52]. Also, NF-κB modulates apical components of metabolic processes including metabolic hormones such as insulin and glucagon, and therefore this pathway is also involved in both insulin resistance and sensitivity, another key component of psoriatic disease-associated MetS [3,49]. The importance between metabolic derangements and NFκB-mediated inflammation associated with psoriatic disease is already moving to the practical realm. Thus, it has recently been set that obese, diabetic, and hypertensive patients with PsA tend to present a better response and persistence to secukinumab, an anti-IL17A therapy [53].

## 6. Conclusions

The NF-κB is an essential regulator of inflammatory and metabolic processes associated with psoriatic disease. Genetic variants of this pathway are associated not only with the risk of suffering the condition but may also be at the base of the comorbidities that frequently accompany it. These variants can also help improve treatment selection. In recent years, genetic variant analysis of the NF-κB pathway has contributed to a finer dissection of the genetic architecture of psoriatic disease. However, we still need further analysis of these genetic variants, especially with an emphasis on non-canonical regulators of this pathway, which currently have a direct implication in the appearance of important comorbidities of psoriatic disease such as diabetes and obesity. Thus, for example, an IKK/TBK1 inhibitor has been tested in obese individuals with type 2 diabetes showing encouraging results [52]. These findings may have future implications for a disease approach focused on the needs of each psoriatic patient.

## Figures and Tables

**Figure 1 ijms-22-13004-f001:**
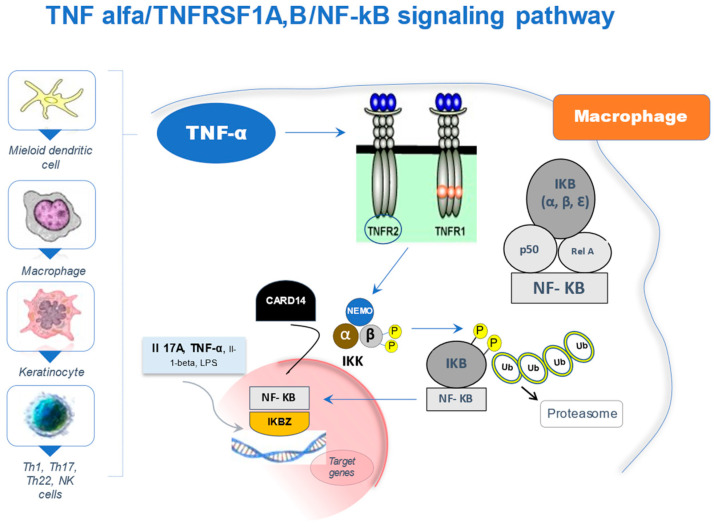
The TNF-α/TNFRSF1A,B/NF-KB signaling pathway. TNF-α is a pro-inflammatory cytokine synthesized by various cell lineages, mainly macrophages, dendritic cells, keratinocytes as well as various T cell types. It exerts its functions through binding to type 1 (TNFR1) and 2 (TNFR2) cell membrane receptors. TNFR2 (TNFRSF1B) shows higher affinity for TNF-α. NF-κB is a dimer made up of Rel family proteins (p50, p52, Rel A/p65, cRel, and Rel B), the p50/p65 heterodimer being the most common. In the absence of TNF-α stimulation, the activity of NF-κB is regulated at cytoplasmic level by an inhibitory complex of proteins called IKB or Kappa Beta inhibitors (IkB alpha, beta, and epsilon) that regulate its function by blocking its translocation to the nucleus. The binding of TNF-α to its receptors triggers the activation of the inhibitor kinases of NF-κB or “IKK complex” consisting of IKK-alpha, IKK-beta and IKK-gamma or NEMO. The characteristic event is the phosphorylation of IKB-alpha by the IKKa/b complex, inducing its ubiquitination and degradation by the proteasome; and thus, allowing the release of NF-κB. Once in the nucleus, NF-κB meets another inhibitory protein called IKBZ. Likewise, IKBZ can act as a transcription factor regulated by IL-17A, IL-1beta, and to a lesser extent, by TNF-α. In turn, IKBZ plays an important role in the development and expansion of Th17 lineages. Finally, CARD14 activates NF-κB by partially known mechanisms (see text for more details), although some authors suggest that CARD14 would activate the IKK complex, consequently regulating the activity of NF-κB, leading to an increase in its transcriptional activity. TNF-α: tumor necrosis factor-alpha. TNFR1 (TNFRSF1A): TNF-α type 1 receptor. TNFR2 (TNFRSF1B): TNF-α type 2 receptor. NF-κB: nuclear factor of kappa light chain enhancer of activated B cells. IKB: kappa-beta inhibitors. IKK: inhibitor kinases of NF-κB. IKBZ: z-inhibitor protein of NF-κB. IL: interleukin. Th: T-helper. CARD14: caspase recruitment domain family member 14.

**Table 1 ijms-22-13004-t001:** Genetic variants within the TNF-α/NF-κB pathway and psoriatic disease features.

Genetic Variant	Disease Feature
*CARD14* rsll652075 (p. Arg820Trp)	- Increased risk of developing psoriasis [16,17,18,19,20]. Good anti-TNFα drug response among CC genotype carriers [40].
*TNFSRF1B* rs1061622 (p. Met196Arg)	- Increased risk of developing psoriasis conditioned by HLA-Cw6 positivity [23]. Worse response to anti-TNFα drugs among G genotype carriers [23].
*NFKB1 ** rs28362491 (−94 of ATTG) *NFKB1* rs7667496/rs28362491	- No apparent association with disease risk and/or disease phenotypes [27]. Increased risk for coronary artery disease [51]. - Type 2 diabetes. Impaired renal function [48].
*NFKBIA* rs7152376 C (complete LD with rs12883343)	- Significant association with PsA [33,34] - Risk of myocardial infarction and several IMIDs [28,29]. - Increased psoriasis risk [30,31,32]. Increased psoriasis risk [30,31,32].
*NFKBIZ* rs7637230 *NFKBIZ* rs3217713 ins/ins *NFKBIZ* rs3217713 del/del	- Increased risk of developing psoriasis [36]. - Increased risk of developing psoriasis conditioned by HLA-Cw6 positivity [37]. Insertion allele more frequent among anti-TNFα non-responders, with a recessive effect [42]. - Early-onset coronary artery disease [50].

* Other variants located at the chromosomal region where *NFKB1* is found have been investigated and related to psoriasis risk, specifically rs1020760 (also associated with a positive family history) and rs1609798 [25,26]. LD: linkage disequilibrium. IMIDs: immune-mediated inflammatory diseases. CARD14: caspase recruitment domain family member 14 gene. TNFSRF1B: Type 2-TNFα receptor gene. NFKB1: NFκB subunit 1 gene. NFKBIA: NFκB alpha-inhibitor gene. NFKBIZ: NFκB zeta-inhibitor gene.

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
