# Peer review of "Genetic Variants of the NF-κB Pathway: Unraveling the Genetic Architecture of Psoriatic Disease"

_ijms, 2021, doi:10.3390/ijms222313004_

Round 1

Reviewer 1 Report

The review article by Queiro et al discusses on genetic variants of NFkB in psoriasis. Overall article is well conceptualized, and written. However at some places it need more elaborations, specially the section where author discuss about NFkB variants in comorbidities associated with psoriasis.

It will be good for the general audience to know about various other comorbidities associated with psoriatic disease. 

Author Response

The review article by Queiro et al discusses on genetic variants of NFkB in psoriasis. Overall article is well conceptualized and written. However, at some places it needs more elaborations, especially the section where author discuss about NFkB variants in comorbidities associated with psoriasis.

It will be good for the general audience to know about various other comorbidities associated with psoriatic disease. 

R/ Thanks for this suggestion. In the new version of the manuscript, we have delved into that part as suggested by the reviewer. You will find these changes highlighted in yellow in the corresponding subsection.

Reviewer 2 Report

The manuscript entitled “Genetic variants of the NF-κB pathway: unraveling the genetic 2 architecture of psoriatic disease” by Queiro et al., is interesting for the people working in the area of psoriasis to better understand the comorbidity.  There are number of studies which described the psoriatic disease, the inflammatory molecules, pathway involved and the inhibitors used.  This review aims to present the broader picture of the authors previous work by Coto-Seura et. al. where they highlighted the importance of NFKBIA gene in psoriatic disease and therefore comprehending the new information’s to explore further. The manuscript is nicely written and the genetic prospective of NF-κB pathways is well described even by including some of the comorbidity conditions.

Minor Comments:

  1. Page 2, Line 61-62; It will be nice to have numbering for the three fundamental network i.e (i), (ii) & (iii). Similarly, Line 87-91 use numbering for the three pathways.
  2. The section describing the “NF-κB pathway variats and drug response” it would be better to tabulate some of the inhibitors effecting NF-κB pathway in psoriatic disease and the comorbid conditions.
  3. It will be very easier to follow the genetic variant and disease feature in detail from literature on citing the references side by side in the Table 1.
  4. In the conclusion section authors can mention additional information for future prospective of the NF-κB genetic variants in correlation with comorbidity in psoriatic disease.

Author Response

The manuscript entitled “Genetic variants of the NF-κB pathway: unraveling the genetic 2 architecture of psoriatic disease” by Queiro et al., is interesting for the people working in the area of psoriasis to better understand the comorbidity.  There are number of studies which described the psoriatic disease, the inflammatory molecules, pathway involved, and the inhibitors used.  This review aims to present the broader picture of the authors previous work by Coto-Segura et. al. where they highlighted the importance of NFKBIA gene in psoriatic disease and therefore comprehending the new information’s to explore further. The manuscript is nicely written and the genetic prospective of NF-κB pathways is well described even by including some of the comorbidity conditions.

Minor Comments:

  1. Page 2, Line 61-62; It will be nice to have numbering for the three fundamental networks i.e (i), (ii) & (iii). Similarly, Line 87-91 use numbering for the three pathways.

R/ Thanks for this suggestion that is now incorporated in the new version. You will see these changes highlighted in yellow.

  1. The section describing the “NF-κB pathway variants and drug response” it would be better to tabulate some of the inhibitors affecting NF-κB pathway in psoriatic disease and the comorbid conditions.

R/ Thanks again. We follow your suggestion. With the permission of the reviewer, rather than introducing a new table, we have included an introductory paragraph to this subsection addressing the point suggested by the reviewer. We hope this change is to your liking.

  1. It will be very easier to follow the genetic variant and disease feature in detail from literature on citing the references side by side in the Table 1.

R/ Nice idea. We added references in table 1 according to your suggestion.

  1. In the conclusion section authors can mention additional information for future prospective of the NF-κB genetic variants in correlation with comorbidity in psoriatic disease.

R/ Good comment. We have delved into the part of comorbidities associated with psoriatic disease. This is highlighted in the new version. See changes in yellow in the corresponding subsections.